# Review of Biological Effects of Acute and Chronic Radiation Exposure on *Caenorhabditis elegans*

**DOI:** 10.3390/cells10081966

**Published:** 2021-08-03

**Authors:** Rabin Dhakal, Mohammad Yosofvand, Mahsa Yavari, Ramzi Abdulrahman, Ryan Schurr, Naima Moustaid-Moussa, Hanna Moussa

**Affiliations:** 1Department of Mechanical Engineering, Texas Tech University, Lubbock, TX 79401, USA; Rabin.Dhakal@ttu.edu (R.D.); Mohammad.Yosofvand@ttu.edu (M.Y.); 2Department of Nutritional Sciences, Texas Tech University, Lubbock, TX 79409, USA; Mahsa.Yavari@ttu.edu (M.Y.); naima.moustaid-moussa@ttu.edu (N.M.-M.); 3Obesity Research Institute, Texas Tech University, Lubbock, TX 79409, USA; 4Medical Center, Department of Radiation Oncology, Texas Tech University, Lubbock, TX 79430, USA; Ramzi.Abdulrahman@ttuhsc.edu; 5Cancer Center, UMC Health System, Lubbock, TX 79430, USA; ryan.schurr@umchealthsystem.com

**Keywords:** ionizing radiation, acute and chronic exposure, dose, biological dosimeter, *Caenorhabditis elegans*

## Abstract

Knowledge regarding complex radiation responses in biological systems can be enhanced using genetically amenable model organisms. In this manuscript, we reviewed the use of the nematode, *Caenorhabditis elegans* (*C. elegans*), as a model organism to investigate radiation’s biological effects. Diverse types of experiments were conducted on *C. elegans*, using acute and chronic exposure to different ionizing radiation types, and to assess various biological responses. These responses differed based on the type and dose of radiation and the chemical substances in which the worms were grown or maintained. A few studies compared responses to various radiation types and doses as well as other environmental exposures. Therefore, this paper focused on the effect of irradiation on *C. elegans*, based on the intensity of the radiation dose and the length of exposure and ways to decrease the effects of ionizing radiation. Moreover, we discussed several studies showing that dietary components such as vitamin A, polyunsaturated fatty acids, and polyphenol-rich food source may promote the resistance of *C. elegans* to ionizing radiation and increase their life span after irradiation.

## 1. Introduction

Humans can be subjected to various types and levels of radiation exposure as they age. In addition to background radiation, humans might receive ionizing radiation (IR) from radiotherapy instruments as part of their medical treatment during their entire life [1]. In these treatments (e.g., Leukemia patient treatment), an absorbed dose of 12 to 15 Gy is delivered to the patient’s body, often given in 8 to 12 fractions in two to three treatments per day over 4–5 days a week for several weeks [2,3]. Moreover, humans at any age may be overexposed to ionizing radiation from nuclear accidents [4]. After Soviet cosmonaut, Yuri Gagarin, first orbited the Earth in a Vostok spacecraft on 12 April 1961, human activities in outer space have expanded tremendously [5]. As of 26 September 2019, a total of 565 people have gone to space, and 12 have walked on the Moon. They have spent more than 29,000 person-days in space, including over 100 person-days of spacewalk [6]. Thus, the number of people and the length of time they spent in space increased in the last century. Therefore, understanding the effects of the space environment on human beings is a priority among the stakeholders working on space exploration. Therefore, it is crucial to understand the effects of such radiation on humans as continuing exposure to IR causes biological damage, leading to severe health effects [7,8]. It is also important to lessen IR’s impact on the quality of life, human health, and life span (aging); thus, ionizing radiation is a critical research topic. The model organism, and nematode, *Caenorhabditis elegans* has been studied to assess the effects of radiation exposure at different levels, such as molecular, cellular, and individual, to determine the implications for human health studies [9].

*Caenorhabditis elegans* are worms, commonly known as *C. elegans* (Figure 1), they fall under phylum Nematoda; are non-hazardous, non-infectious, and free-living organisms that live in decayed vegetation (soil), where they thrive by feeding on microbes [10]. *C. elegans* is an ‘in vivo’ model system that is easy to culture in the laboratory because their adults are small, about 1 mm in length [11]. *C. elegans* consists of many sections similar both anatomically and genomically to those found in humans, such as muscles, an integrated nervous system, gut, and reproductive system, and fat body. This nematode is one of the best-suited model organisms widely used to investigate the exposure to different types of ionizing and non-ionizing radiation [12]. *C. elegans* consists of about 20,000 predicted protein-encoding genes, over third of which are homologous to human genes [13], and shares similarities in response to DNA damage response with humans [14].

*C. elegans* model organism has been an excellent model to conduct microbiological experiments to research apoptosis, muscle atrophy, radiation effects, metabolic diseases, and aging, given its short life span compared to humans and other mammals [15]. It has also been used to study embryogenesis, morphogenesis, nerve function, and behavior [16]. In 1976, Herman et al. conducted a remarkable investigation on ionizing radiation’s effects on *C. elegans*. He analyzed the chromosome rearrangement due to irradiation with X-ray to produce a system called balancing lethal mutations using chromosome balancers [17]. Subsequently, in 1985, by irradiating the worm with ionizing radiation, Hartman found various mutants of *C. elegans*; two of them were radiosensitive mutants, which he named rad-1 and rad-2 [18]. Later, the interest in radiation effects on *C. elegans* expanded to various other fields of research such as the DNA damage induced by radiation, DNA repair, aging, behavior, the effects of different types of radiation, the effects of the quality of radiation, etc. [12,19,20]. Here, we provide a review of current experiments on *C. elegans* with different types, doses, and radiation levels with a special focus on irradiation effects based on the dose intensity and exposure times. This paper will be useful for further studies of *C. elegans* with a different intensity and irradiation time.

## 2. Life Cycle

The life span of *C. elegans* is influenced by temperature [23,24], and they live for 23 days at 16 °C and nine days at 25 °C [25]. The average life span of *C. elegans* is 2–3 weeks and the generation time (from egg to adult) is about 56 h at 20 °C; the adults are about 1 mm in length. Figure 2 illustrates the life cycle of *C. elegans* (hermaphrodite). Similar to that of other nematodes, the life cycle of *C. elegans* consists of the embryonic stage, four larval stages (L_1_–L_4_), and adulthood [26]. Each larval stage ends in a molting stage, during which a new, stage-specific cuticle is grown, and the old cuticle is shed [27]. The size of the animal increases throughout the four larval stages, but distinguishing between separate sexes is not easy until the L_4_ stage [22]; at the L_4_ stage, hermaphrodites have a tapered tail, and the developing vulva can be seen in the middle of the ventral side as a transparent half-circle. At this point, the broader tail is seen on males but without a distinctive fan (later young adult consist of fan-shaped tail).

*C. elegans* can take an alternative form of life, called a dauer larval stage, if the plates are too crowded or if food is scarce [28]. Dauer larvae are thin and mobile, but they cannot eat; interestingly, dauers can remain viable for up to four months [29]. They appear non-aging: dauer larvae can roam around for months and then re-enter the L_4_ stage when they encounter a food source and live about 15 more days. *C. elegans* can live nearly ten times their average life span [30]. At 25 °C, a generation time is about 3–4 days, whereas it extends to almost a week at 16 °C; some mutations in *C. elegans* are also temperature sensitive, only showing a phenotype at either 16 or 25 °C [31].

### 2.1. Factors Affecting the Life Span of C. elegans

The life cycle duration for *C. elegans* depends on various factors such as diet, temperature, reproduction, and oxygen level [32]. Additionally, the effect of irradiation on *C. elegans* varies according to the treatments during the experiments. Herein, we review the different kinds of treatments performed on *C. elegans* before, during, and after the radiation exposure and discuss the worm’s biological response.

#### 2.1.1. Temperature

Although the standard temperature of growing *C. elegans* in a laboratory is 25 °C, studies are often conducted under different temperatures within the worm physiological range, i.e., 15 to 20 °C [33]. Therefore, this variation of temperature would modify the mean life span, and also, the worms experiencing low temperature during development experience a short life span [34]. When grown in a liquid medium, the mean life span of *C. elegans* is found to be 23 days at 16 °C, but only nine days at 25 °C [25]. However, mutants of *C. elegans* defective in clk (clock) genes (involved in regulation of circadian rhythm) cannot adapt their growth rate to temperature changes [35].

#### 2.1.2. Diet

Growth is significantly disrupted if larva hatches in the absence of food; however, they usually grow into adulthood when they feed, and their later adult life span is the same as that of a worm that hatches in the presence of food [36]. The life span of *C. elegans* can be significantly extended by calorie restriction [24,37,38]; it can be increased by around 50 percent for *C. elegans* grown in liquid cultures with a relatively low amount of feeding bacteria [25]. Moreover, *Comamonas* sp. and *E. coli* strain HB101 are the optimal food for the nematode, whereas Bacillus *megaterium* is the worse food. *E. coli* DA837 and *Bacillus* simplex are intermediate food sources [39]. Additionally, several investigations reported that food sources that contain phytochemicals such as blueberry polyphenols [40], epigallocatechin gallate from tea [41], plant adaptogens [42], and tart cherry [43], among others, increase the life span of *C. elegans*.

#### 2.1.3. Reproduction

An inherent relationship exists between the reproductive state and the life span of *C. elegans* [44,45]. The life span of wild-type *C. elegans* could be increased with the loss of germline cells [46,47,48]. Male life span appears shortened by mating or mating attempts, but the hermaphrodite life span is not affected [49]. However, a more extensive study found that mating with males reduces the hermaphrodite life span by up to one-half [50].

#### 2.1.4. Oxygen Level

The level of oxygen impacts *C. elegans*’ life span. *C. elegans* grown in high oxygen concentrations have a shorter life span than those grown in low oxygen concentrations [51,52], which can be justified with the finding that hypoxia interrupts proteostasis [53]. Moreover, *C. elegans* have developed the ability to survive at least 48 h in a state of anoxia [54].

## 3. Acute and Chronic Irradiation in *C. elegans*

Research on *C. elegans* irradiation can be divided into the following two types based on the intensity and the length of exposure: acute and chronic exposures. Acute irradiation is a short-term exposure to a high dose, while chronic irradiation is prolonged exposure to a low dose over a determined period of time [55]. Quantitatively, exposures greater than 0.1 Gy/min would be accepted as acute, while exposures of less than 1 Gy/h would be accepted as chronic exposures by most radiobiologists; however, the intermediate levels are not well defined [56]. In nature, most of the irradiation from the solar particle events is characterized as acute irradiation, whereas irradiation from galactic cosmic rays is characterized as chronic irradiation (Figure 3) [57]. In addition to the natural irradiation of *C. elegans* that were flown to space for experiments, various other ground-simulated acute and chronic irradiation of *C. elegans* are discussed in this section. Both types of irradiation decreases the life span, increases the death ratio compared with control groups, and cause severe DNA damage, leading to genetic mutations and less fertility in *C. elegans*. The irradiation from solar particle events is short-term but is of high intensity; hence, it is considered that it is acute irradiation that has a high impact on the *C. elegans*’ biological life causing increased in the death and DNA damage and a decrease in life span, genetic mutation, and less reproduction [57]. The irradiation from galactic cosmic rays is considered a low dose, but a high exposure time; hence, it is considered chronic irradiation and it changes the biology of *C elegans* less than acute irradiation relatively [58].

After the original study evaluating the effect of ionizing radiation on *C. elegans* by Herman et al. in 1976 [17], much work has been conducted on the topic of acute and chronic radiation exposure. Johnson et al. exposed a wild-type and radiation-sensitive mutant to gamma rates of 0.027 Gy/min from a Cesium radionuclide (137Cs) at different developmental stages of *C. elegans* to monitor the life span; they found that at least 0.1 Gy is needed to reduce the mean life of *C. elegans* [59]. Moreover, they have also found the same conclusion drawn by Johnson et al., that dauers are the most sensitive, which may be due to arrest of development, while eight-day-old adults are the most resistant to the ionizing radiation [36,59]. Similarly, various other studies focusing on different conditions are discussed and also a summary of the irradiation with various types of ionizing radiation is presented in Table 1 and Table 2.

### 3.1. Space vs. Ground Experiments

The number of mutations observed in *C. elegans* due to the exposure to naturally occurring radiation in space is proportional to the exposure time. This enabled the *C. elegans* to be used as a biological dosimeter, as shown in various space experiments. For example, G. A. Nelson et al. observed the development of *C. elegans* as a function of gravity and space radiation exposure through a study called Genetic Molecular Dosimetry of HZE Radiation, where *C. elegans* were prepared and kept inside the International Microgravity Laboratory of the Discovery Space Shuttle were flown on 22 January 1994 for about eight days in space [60]. The physical dosimetry was performed with thermoluminescence dosimetry (TLD), a versatile passive dosimeter that recorded a total dose of 0.8 to 1.1 mGy in the Spacelab tunnel where protection against radiation is minimal. This research concluded that the radiation-induced mutation rate in *C. elegans* in space is eight times greater than in the ground control group, demonstrating a significant risk of cancer inherent in extended space travel [60].

Another experiment in space assessing the effects of space radiation on the biological growth of *C. elegans* has been extensively studied by Hartman et al. [61]. The fem-3 gene of *C. elegans* was investigated to find the mutation frequency and the nature of mutations caused by space radiation in low earth orbit during Space Shuttle flight STS-76. The Space Shuttle was flown with *C. elegans* on 22 March 1996, and the radiation exposure time was about nine days. Dauer larvae were screened for mutants; a total of 25 fem-3 mutants were recovered and yielded a mutation frequency of 2.1 × 10^–5^, which is approximately 3.3 times higher than the spontaneous rate of 6.3 × 10^–6^. A radiation dose of 0.268–0.306 cGy was measured during the space flight with a dose equivalent to 0.592–0.892 cGy and an average quality factor of 2.59. The mutation frequency was found to be significantly increased with the increased radiation exposure. The presence of a significant number of putative deficiencies in the recovered genes of *C. elegans* suggests that charged iron particles are the major mutagenic component, and the increased mutation frequency brings a significant cancer risk in extended space travel [61]. Thus, NASA began research on *C. elegans* as a biological dosimeter in 2004, based on the findings of previous experiments by G. A. Nelson et al. and Phil S. Hartman et al., which showed that the number of mutations observed in *C. elegans* is proportional to the amount of time exposed to naturally occurring radiation in space [62]. The agency launched the International *Caenorhabditis elegans*-First Flight (ICE-first), where *C. elegans* was used as a model organism to investigate the biological effects of a short duration (about 11 days) spaceflight [62]. The research focused on studying the mutational effects of spaceflight, where they utilized a genetic balancer system known as eT1 for capturing, maintaining, and recovering mutational events that occurred over several generations during spaceflight. During this experiment, *C. elegans* were grown for the first time in defined liquid media during the spaceflight. This research concluded that there was no significant difference in the mutation rates during the short space flight. The researchers demonstrated that the T1 balancer system can be used for longer-term biological damage measurements, as the model system was relatively simple and robust and it also could be used as a biological dosimeter [63].

Besides the space-based experiments, numerous simulated ground-based radiation tests demonstrated the same changes observed during exposure in space. Yi et al. performed a ground simulation on *C. elegans* where they simulated a long-duration flight and analyzed the effect of space radiation and gravitational force (G-force) on *C. elegans* [64]. In their experiment, they exposed *C. elegans* to accelerated protons and gamma rays with an equivalent dose. During long space flights of different durations, the anticipated radiation dose was calculated based on Reitz simulations and methods. They simulated the dose received by *C. elegans* during space flights lasting one month, six months, and two years and gave different doses to *C. elegans*. They analyzed whole-genome using microarray and investigated the worm phenotypes; the results of their experiment showed that protons and gamma rays provoked genetic changes related to response to DNA damage, oxidative stress, and cell death [65]. These exposures to ionizing radiation regulate (provoke or reduce) embryonic development ending in egg hatching and birth. This study predicted that accelerated protons induce a gene expression that is related to the response to DNA damage and anti-apoptosis, whereas gamma rays induce apoptosis.

Guo et al. studied the effect of a radiation-induced bystander in cells not directly irradiated in *C. elegans* [66]. Worms were grown in room conditions and were fed with standard E. coli OP50. Proton radiation was used in these experiments, and the dose delivered by each proton hitting the *C. elegans* was calculated as 0.033 Gy. The L_4_ stage worms were irradiated by proton beams with 1.65, 6.6, 16.5, 33, and 66 Gy and were compared with the control group. The number of dead germ cells was counted 24 h after the irradiation. The results clearly showed that bystander effects happen in *C. elegans* after ionizing irradiation, including effects on germ cells [66]. In a similar study, Tang et al. used a combination of irradiation with a proton beam and gamma rays to investigate the interaction between the radio-adaptive response and the radiation-induced bystander effect (RIBE) in *C. elegans* [67]. In this experiment, they performed whole-body exposure of a 14-h-old *C. elegans* to a microbeam of 1000 and 2000 protons, respectively. After two hours, the worms were irradiated with gamma radiation with 75 and 100 Gy, respectively. Their study suggested the phenomenon of RIBE, where defects such as the vulva abnormalities (non-irradiated cell) can be significantly decreased because of a signal received from the irradiated rectal valves (irradiated cell) of *C. elegans*; therefore, the radio-adaptive response of the *C. elegans* can be improved by a radiation-induced bystander effect. Xu et al. exposed *C. elegans* to X-rays to study the response of *C. elegans* to ionizing radiation [68]. In this study, the following three *C. elegans* cohorts were included: a control group and two groups exposed to X-ray doses of 200 and 400 Gy. After irradiation, total RNA was extracted from each group, also RNA was sequenced to compare the transcriptomes between these groups. They found that many genes expressed differentially and were enriched significantly in ionizing radiation-related biological pathways and processes [68]. This demonstrated that *C. elegans* genes related to various biological processes, such as behavior, regulation of growth and locomotion, positive regulation of growth, calcium ion transport, and di- and trivalent inorganic cation transport, were modulated by different doses of X-ray radiation [68]. Ionizing radiation affected the locomotion of *C. elegans* and influences the reaction of *C. elegans* to the environment. Sakashita et al. studied the effects of dietary NaCl on the learning (learning function of nervous system) of radiation-induced *C. elegans* [69]. The *C. elegans* were grown at room condition and fed with E coli until the adult stage. The *C. elegans* were exposed to both acute and chronic irradiation in two separate groups. The animals were irradiated with a 100 Gy dose of gamma-ray at the rate of 32 Gy/min as acute irradiation. In chronic irradiation, the animals were also irradiated with a 100 Gy dose of gamma-ray but at a rate of 0.42 Gy/min for four hours. The animals were then treated in the following three sets of NGM plates: (1) without NaCl, (2) 10 Mg NaCl, and (3) 50 mM NaCl to measure the chemotaxis toward NaCl. Their experiment showed that both acute and chronic irradiation decreases the chemotaxis of *C. elegans* to NaCl. They suggested that irradiation in *C. elegans* may have a modulatory effect on diet NaCl associated with animals learning.

### 3.2. Effect on Aging

The effects of exposing *C. elegans* to ionizing radiation are a subject of interest in radiation biology research investigating the process of aging in this organism as the life span of *C. elegans* is affected by the radiation exposure. This was investigated in a study by Kuzmic et al., where they show an acceleration of the aging process by exposing *C. elegans* to chronic irradiation. Irrespective of the dose rate and dose delivered, chronic exposure reduced the level of oxidative protein damage (carbonylation), which is a biomarker of the aging process [70]; Dubois et al. also experimented with the response of *C. elegans* to acute and chronic exposure [71]. After chronic exposure, 168 proteins were significantly changed in the experimental group, while 369 proteins were significantly changed following acute radiation exposure. Analysis of global protein expression in this study showed the modulation of proteins involved in regulating biological processes such as lipid transport, DNA replication, development of germ cells, apoptosis, transport of ions, and the development of cuticles. These results confirmed that chronic irradiation induced different molecular mechanisms than acute irradiation [71].

Besides the whole-body irradiation of *C. elegans*, partial irradiation is becoming more important in recent studies. Suzuki et al. irradiated specific regions on the *C. elegans* to compare animal movements with whole-body irradiation [72]. *C. elegans* were grown at 20 °C and were fed with *E coli*. Then, the three-day-old animals were irradiated with a 500 Gy dose of heavy carbon ions. The worms were irradiated within four groups. The groups were as follows: whole-body irradiation, central nervous system, CNS, irradiation or head, middle of the body irradiation, and tail irradiation. They also used active animals in their experiment and studied the motion of the animals immediately after the irradiation, which was the novelty of their experiment method. The body bend rate was calculated for five animals in each group. The results showed that the whole body irradiation reduces the mobility of the worms, while region-specific irradiation did not significantly affect the motion of the *C. elegans* [72]. Another study by Suzuki et al. presents evidence of the CNS’s activity in reducing the mortality in adult hermaphrodite *C. elegans* due to acute irradiation. Research shows that decreased mortality after irradiation of the CNS depends on the dose, and the body muscle around the CNS is also partly involved in reducing the mortality [73]. A device that makes the researcher able to partially irradiate *C. elegans* has recently been developed by Funayama et al. [74]. They have designed and built a device to irradiate animal models such as *C. elegans* with an ionized microbeam. They obtained a beam size smaller than 10 μm in their device to target specific cells in their animals under the microscope. They irradiated intracellular objects with the newly developed device. They grew *C. elegans* in a room temperature environment at 20 °C for three days, and the animals were cultured on nematode growth medium [74]. The CNS of the *C. elegans* was irradiated with 500 Gy of carbon ions. The microbeam’s size was 8 μm, which helped to irradiate the worms’ nervous system spot. Following irradiation, samples were collected, and the animals were placed on an NGM plate where they were not fed anymore. Then, they evaluated the effects of the irradiation on the locomotion of the *C. elegans*. The results showed that the irradiation did not change the animals’ locomotory rate, and thus, the 500 Gy had no effect on the movement of the *C. elegans*. They also designed a frame in which they can irradiate the *C. elegans* with micrometer precision, which can be used for further investigations on the heavy ion irradiation on *C. elegans* and cultured cells [74].

### 3.3. Effect on Reproduction

Radiation exposure affects *C. elegans* in its reproduction. Min et al. showed that *C. elegans* irradiated using a proton beam at the L_4_ larval stage grew almost generally in the adult stage, but later experienced reproductive deficiencies demonstrated by the decreased size of brood as measured by the total number of eggs that are fertilized by hermaphrodite animals; moreover, an increase in germline apoptosis caused reduced fertility [75]. The effects of a heavy-ionized beam on *C. elegans* germline cells were studied by Sugimoto et al. [76]. The nematodes were grown in room conditions at 20 °C for 24 h on NGM plates. When the animals entered the L_4_ larval stage, they were exposed to 20, 40, 60, and 100 Gy doses of carbon ion microbeams for both whole-body irradiation and germline cells. Then, the nematodes’ hatching rate was calculated at 0 to 4 h and 11 to 15 h after the irradiation. The hatching rate for the delivered dose before 4 h was 65, 20, 2, and 0 percent, respectively for the surviving eggs. The calculated rate for 11 to 15 h was about 100, 90, 60, and 0 percentages for the surviving eggs. The experiment results showed that both whole-body irradiation and germline spot irradiation decrease the hatching rate of the larva [76]. Histone-modifying enzymes have been reported to affect longevity through several generations [77,78]. Their progeny may gain trans-generationally inheritable survival benefits [79].

Min H et al. reported major increases in the number of proteins named RAD-51 and HUS-1 (RAD-51 and HUS-1 are both critical components in the DNA damage response of *C. elegans*) and in the number of chromosomal aberrations in gonads and oocytes, respectively, following proton beam irradiation in hermaphrodites P0. However, these defects were recovered mainly in progeny F1 and F2 [75]. Homologous recombination resects the double-strand breaks (DSBs) and produces single-stranded DNA tails. Since RAD-51 binds directly to these single-stranded DNA tails, a good estimate of the number of DSBs in a cell is given by the number of these proteins [80]. An increase in the number of RAD-51 foci (6.0 ± 1.75 per nucleus) in hermaphrodite gonad germ cells relative to those of non-irradiated hermaphrodite gonad germ cells (0.9 ± 0.61 per nucleus) was observed following irradiation at a dose of 10 Gy with a proton beam [80]. A critical component of the response to DNA damage in *C. elegans* is HUS-1; compared to non-irradiated gonads, the number of HUS-1 proteins was found to have increased in the irradiated hermaphrodite gonads [81]. Ryu et al. studied the roles of a homolog of p53 binding protein 1, HSR-9, in the DNA repair and the ionized irradiation response of *C. elegans* [82]. Standard *C. elegans* strains were cultured in room conditions and fed with E. coli. Furthermore, the worms were grown until the L_1_ stage, and then the worms were used for different experiments such as survival gem cells rate and abnormality developments in the eggs of the next generation of worms. Twenty different RNA-types of L_4_ stage animals were irradiated with 60 and 120 Gy doses of gamma-ray to find the germ cell survival of *C. elegans*. The hatching rate for the eggs was calculated at 24 and 48 h after the irradiation. Additionally, the survival rate for L_1_ stage worms was calculated in a separate experiment [82]. The nematodes were exposed to 25 and 50 Gy doses of acute gamma irradiation, and the hatching rate was calculated after 48 h. Furthermore, the 4-h-old eggs of *C. elegans* were irradiated with a 90 Gy dose of Gamma-ray searching for any abnormalities in the hatched worms after the irradiation. Both the irradiated and the control group of worms were examined until the L4 stage and abnormalities were recorded in the irradiated worms after three days. The results of these experiments showed that although p53 binding protein one homolog increases apoptosis, it does not directly influence the *C. elegans* response to DNA damage [82].

### 3.4. Effect on DNA Response

The DNA response to ionizing irradiation in *C. elegans* has gained much interest recently. Bertolini et al. studied DNA repair mechanism in irradiated *C. elegans* [83]. They used wild-type *C. elegans* and compared their DNA response to other mutant *C. elegans* strains, such as brc-1, bub-3, and san-1. The *C. elegans* were grown in room temperature conditions and fed E coli OP-50 and irradiated using a 137Cs gamma source. The L_1_ stage worms were exposed to a 60 Gy dose of ionizing irradiation, which resulted in vast DNA damage in *C. elegans*, but it did not decrease the worms’ reproduction. Other *C. elegans* were irradiated with 90 and 120 Gy doses, and results were compared to a control non irradiated group [83]. They found that the lethality increased in the bub-3 mutant compared to wild-type *C. elegans*. Additionally, ionized sensitivity in both bub-3 and san-1 was increased. These studies suggested that the involvement of bub-3 in response to the DNA damage in *C. elegans* occurs in cell cycle timing [83]. In another genetic study, Vujin et al. exposed different mutants of *C. elegans*, as well as wild-type N2 and wild-type NHJ1 strains to ionizing radiation [84]. The *C. elegans* were irradiated by 0, 25, 37.5, 50, and 75 Gy doses of X-ray in different experiments and different factors, such as ionizing radiation sensitivity or resistance, were determined. They suggested that damage in the nhj-1 sequence caused sensitivity to ionized radiation in wild-type N2 *C. elegans* [84]. Yang et al. showed ceramide regulated DNA damage response to mediate radiation induced germ cell apoptosis [85]. Buisset-Goussen et al. described a decrease in reproductive capacity in future generations after chronic exposure of *C. elegans* mother hermaphrodites to gamma radiation [86]. Barber et al. reported that the genomic instability in the progeny, which was not irradiated, was caused by irradiation of the mother *C. elegans* and concluded that the damage due to radiation is a long-term risk that lasts for generations [87]. Proton beam irradiation was also reported to cause transgenerational effects in three successive generations by Min et al. [75].

In another experiment, *C. elegans* were exposed to both chronic and acute γ-ray radiation in two different experiments by Dubois et al. [88]. They showed that acute irradiation influences the hatching of the worms, but chronic irradiation does not. *C. elegans* were exposed to six different doses and four doses in acute and chronic irradiation experiments. *C. elegans* were irradiated for 65 h during a whole life cycle from the embryo stage to the L4-YA adult stage for chronic exposure. The study’s radiation source was 137Cs, which provided a homogeneous dose rate on the plate [88]. A similar study by Maremonti et al. about the acute and chronic exposure of *C. elegans* to γ-ray radiation demonstrated that toxicity of chronic exposure in reproduction compared to acute exposure [89]. While acute irradiation did not significantly affect reproduction, chronic irradiation of 62 h from egg to young adult stage decreased the number of larvae hatched per adult hermaphrodite *C. elegans* up to 61% with a total dose of 14 Gy. These effects were mainly due to increased germ cell apoptosis, impaired sperm meiosis, with adverse effects on sperm production [89]. Later studies by Maremonti et al. showed an increase in the copy of mitochondrial DNA (mtDNA), which plays an important role in reproduction influencing germ cell development, quality, and embryonic development after chronic exposure [90]. Moreover, in their latest study, Meremonti et al. investigated effects of chronic exposure of gamma radiation on accumulation of free radical and the subsequent antioxidant responses to the apical reproductive and developmental processes in *C. elegans*. This research showed a reduction in the number of offspring by 20 and 40% after 96 h of gamma exposure of rate 40 100 mGy h^−1^ (total dose ~3.9) and 100 mGy h^−1^ (total doses ~9.6 Gy), respectively [91].

**Table 1 cells-10-01966-t001:** Summary of Acute Irradiation of *C. elegans*.

Type of Radiation and Exposure Time	Radiation Dose/Dose Rate	Findings	Year/ Studies
Gamma Radiation for about an hour	0.027 Gy/min	≥0.1 Gy is needed to reduce the mean life of *C. elegans*. Dauers are the most sensitive and 8-day-old adults are the most resistant to ionizing radiation.	Johnson et al., 1984 [36]
Targeted micro bream (12C ion particles)	20, 40, 60, and 100 Gy	Reproduction in *C. elegans* eggs arrested for both whole body and tip irradiation. In spot irradiation, the neighboring cells around the targeted point did not arrest the reproduction of germ cells nor did apoptosis happen.	Sugimoto et al., 2006 [76]
Gamma radiation	100 Gy (32 Gy/min)	Chemotaxis reaction of *C. elegans* toward NaCl decreases. The role of ionizing irradiation in associative learning of *C. elegans* toward NaCl is elaborated.	Sakashita et al., 2008 [69]
Gamma radiation for 18 s	6 × 10^−3^ to 2.8 × 10^−2^ Gy (for 1 month) 36 × 10^–3^ to 16.8 × 10^–2^ Gy (for 6 months) 144 × 10^–3^ to 67.2 × 10^–2^ Gy (for 2 years)	*C. elegans* would experience some damage from irradiation during long-term space flight, there are changes in genes related to DNA damage response, oxidative stress, and cell death, and the gamma rays induce apoptosis.	S. Yi et al., 2013 [64]
Accelerated proton for 18 s	33.6 × 10^–3^ to 16.8 × 10^–2^ Gy (for 6 months) 144 × 10^–3^ Gy (for 2 years)	DNA repair mechanism was reduced due to proton exposure. Accelerated protons induce the expression of genes that are related to the DNA damage response and anti-apoptosis.	S. Yi et al., 2013 [64]
Gamma radiation	25, 50, 60, 90, 120 Gy	53BP1 homolog, HSR-9 increases the cell death in muted *C. elegans* exposed to acute irradiation. HSR-9 does not involve in *C. elegans* cells’ response to DNA damage due to ionizing irradiation.	Ryu et al., 2013 [82]
Proton particles microbeam	1.65, 6.6, 16.5, 33 and 66 Gy (0.033 Gy per proton particle)	Proton irradiation increases the germ cell apoptosis in neighboring cells around the radiation spot. Ionized irradiation causes bystander effects in *C. elegans* cells	Guo et al., 2013 [66]
Proton microbeam and gamma rays (137Cs) (Time not specified)	Proton bream: 3.2 MeV with linear energy transfer rate Gamma rays: 75 and 100 Gy	DNA damage in worms was unique and in somatic cells, which include vulval cells the DNA damage checkpoint was not active. Radio-adaptive responses of the whole *C. elegans* organisms were improved by bystander effect, which was induced by radiation.	Tang et al., 2016 [67]
Targeted micro bream (12C ion particles)	500 Gy	Development of a method to irradiate active *C. elegans*. Whole-body irradiation decreased the movement rate of *C. elegans* significantly. Regional irradiation on the head, middle, and tail of *C. elegans* did not have a significant effect on the movement rate.	Suzuki et al., 2017 [72]
Gamma radiation (137Cs)	0, 60, 90, and 120 Gy	Ionized irradiation caused significant damage in *C. elegans* DNA but did not reduce the reproduction cells. Sensitivity to ionized irradiation increased in *C. elegans* mutants compared to the wild-type strain. BUB has a role in the response of *C. elegans* to DNA damage.	Bertolini et al., 2017 [83]
Gamma radiation (137Cs)	2.5, 6.5, 14.4, 50, 100, and 200 Gy	Dependency of hatchability on irradiation dose was shown by the result of the decrease in significant number of progenies per individual after irradiation from and above 50 Gy, until 200 Gy.	Dubois et al., 2018 [88]
Gamma radiation (137Cs)	1 Gy·min^−^^1^ 0.5 Gy, 1 Gy, and 3.3 Gy	369 proteins were found with significant differences. The molecular mechanisms induced by chronic irradiation differ from those induced by acute irradiation.	Dubois et al., 2019 [71]
X-ray (600C/D linear accelerator)	0 gray [64], 200 Gy, and 400 Gy (not specified)	Genes related to several biological processes, such as behavior, regulation growth and locomotion, positive regulation of growth, calcium ion transport, and di- and trivalent inorganic cation transport, are differentially expressed	Xu et al., 2019 [68]
Gamma Radiation	Dose Rate: 1445 mGy·h^−^^1^ Total dose up to 6 Gy	Acute irradiation does not induce a significant change in reproduction.	Maremonti et al., 2019 [89]
Targeted micro bream (12C ion particles)	0, 500, 100, and 1500 Gy	The decrease in mortality depends on the dose due to central nervous system (CNS) targeted irradiation and may partly be due to body-wall muscle cells around the CNS.	Suzuki et al., 2020 [73]
Targeted micro bream (12C ion particles)	500 Gy	Targeted heavy ion microbeam smaller than 10 µm. The preparation and irradiation method for the device is provided. Targeted irradiation on the specific spots did not have an impact on the movement of *C. elegans*.	Funayama et al., 2020 [74]
Gamma Radiation	0, 5, 10, 25, and 50 Gy (3.37 Gy/min)	Germ cell apoptosis decreases when *C. elegans* are treated with Ceramide. Ceramide influences *C. elegans*’s response to DNA damage. Ceramide involves in the functioning of mitochondria in *C. elegans* under ionizing irradiation.	Yang et al., 2020 [85]
X-ray	0, 25, 37.5, 50, and 75 Gy	NHEJ factor in *C. elegans* is reported. NHJ-1 causes ionized radiation sensitivity in N2 wild-type *C. elegans*.	Vujin et al., 2020 [84]

**Table 2 cells-10-01966-t002:** Summary of Chronic Irradiation of *C. elegans*.

Type of Radiation & Exposure Time	Radiation Dose	Findings	Year/Studies
Low dose gamma-ray radiation for 219.5 h	0.268 to 0.306 cGy	The mutation frequency increased significantly due to exposure to space radiation. The charged iron particles are the major mutagenic component and the increased mutation frequencies caused significant cancer risk inherent in extended space travel.	Hartman et al., 2001 [61]
Low dose gamma-ray radiation for 11 days	Not specified	No significant increase in the mutation rate Introduction of eT1 balancer system for longer-term measurement of biological damage in space.	Zhao et al., 2006 [62]
Gamma radiation for 4 h	100 Gy (0.42 Gy/min)	The avoidance response of *C. elegans* toward NaCl decreased significantly.	Sakashita et al., 2008 [69]
Accelerated proton for 18 s	33.6 × 10^–3^ to 16.8 × 10^–2^ Gy (for 6 months) 144 × 10^–3^ Gy (for 2 years)	DNA repair mechanism was reduced due to proton exposure. Accelerated protons induce the expression of genes that are related to the DNA damage response and anti-apoptosis.	Yi et al., 2013 [64]
Gamma Radiation (137Cs source) for 64 h	Dose rate 7 and 52 mGy/h Dose: 0.5 and 3.3 Gy	Life span is significantly shortened in irradiated *C. elegans*. There was a significant difference between different absorbed doses for the same dose rate.	Kuzmic et al., 2019 [70]
Gamma Radiation (137Cs source) for 19 days	Dose rate 7 and 52 mGy/h Dose: 3.3 and 24 Gy	Life span is significantly shortened in irradiated *C. elegans*. There was a significant difference in absorbed doses in the treatments between 3.3 Gy cumulative irradiation (with 7 mGy/h) and 24 Gy cumulative irradiation (with 52 mGy/h)	Kuzmic et al., 2019 [70]
Gamma Radiation (137Cs source) for 65 h	Six dose rates 7, 14, 45, 50, 75, and 100 mGy/h Dose: 0.5, 1, 3.3 Gy	There are no effects from irradiation on the percentage of the hatch after chronic irradiation compared to control *C. elegans*.	Dubois et al., 2018 [88]
Gamma radiation (137Cs source) for 65 h	Dose rate: 7, 14, 50 mGy·h−1 corresponding to cumulated doses (0.5, 1, and 3.3 Gy)	168 proteins were found with significant differences. The molecular mechanisms induced by chronic irradiation differ from those that were induced by acute irradiation.	Dubois et al., 2019 [71]
Gamma Radiation for 62 h	Dose rate: 0.9 to 227 mGy·h−1 Total dose up to 228 Gy	The number of larvae hatched was significantly decreased (by 43 and 61%, when chronically exposed from egg to young adult stage to a total dose of 6.7 Gy and 14 Gy, respectively) with increased germ cell apoptosis, impaired sperm meiosis, and adverse effects on sperm production.	Maremonti et al., 2019 [90]
Gamma Radiation for 72 h	Dose rate: 0.4 to 100 mGy Dose: 0.03 to 72	Significant increase in mtDNA copy number (approx. 1.6-fold).	Maremonti et al., 2019 [90]
Gamma Radiation for 96 h	Dose rate: 40 and 100 mGy·h_−1_ Total dose: ~3.9 and 9.6 Gy	Toxic effect in reproduction. No of offspring reduced by 20 and 40%.	Maremonti et al., 2020 [91]

Besides these effects on the various processes of the life cycle of *C. elegans*, there are several other important radiation studies that have been conducted. Weidhaas et al. developed a tissue model of reproductive cell death induced by radiation in *C. elegans* [92]. Huangqui et al. demonstrated the radio-adaptive response for reproductive cell death in the valval tissue in *C. elegans* for the first time and showed that the *C. elegans* is an excellent in vivo model for radiation-induced reproductive cell death [93]. Similarly, Liangwen et al. demonstrated reproductive cell death across the entire range of carbon-ion irradiation [94] These studies opened new horizons on radiation-induced reproductive cell death in *C. elegans* and further confirmed that this nematode is an excellent model to study human reproductive cell death from the radiation therapy that is used in cancer therapy.

It can be inferred from the study of Table 1 and Table 2 that there is a variation of biological responses by the *C. elegans* irradiated with different radiation types and dose rates. For a similar type of radiation, it is obvious that the higher dose of radiation affected biological responses more than that the lower dose did. Moreover, there is distinctive variation of biological responses by irradiation from low-LET and high-LET (LET- linear energy transfer). High-LETs are highly mutagenic compared to low-LETs to diploid animal cells in an in vivo based system such as the *C. elegans* model organism. On the other hand, biological response are also affected by the treatment of *C. elegans* and the environment where they were grown, before and during the irradiation. Despite all these conditions, *C. elegans* has been regarded as an excellent model organism for radiation biology research. However, there are also some disadvantages or limitation of the *C. elegans* model system. First, *C. elegans*, having a simple body plan, lack many defined organs or tissue such as brain, blood, defined fat tissue or cells, and internal organs found in humans. Second, *C. elegans* has a very short length of about 1mm, which make it difficult to conduct some biochemical experiments such as understanding tissue-specific signaling [95].

## 4. Dietary or Pharmacological Interventions to Reduce Effects of Radiation

Recently, certain phytochemical components have been used to improve *C. elegans*’ resistance against ionizing irradiation. The protective effects of Eleutheroside E (EE) against gamma irradiation have been studied by Liu et al. [96]. Eleutheroside E is a compound form Acanthopanax senticosus, which is a native plant of East Asia, found in China and Japan. *C. elegans* were grown in normal room conditions, and L_3_–L_4_ larvae were used in the experiment. The animals were fed with EE 24 h before and then 24 h after the irradiation. *C. elegans* were irradiated with 50 Gy of gamma-ray at the rate of 0.33 Gy/min for 2.5 h, and then the head thrashes and body movements were measured. The results showed that EE significantly protected *C. elegans* against the damage caused by gamma irradiation. They also showed that the long-term memory of the *C. elegans* induced by gamma irradiation was improved after treatment with EE [96]. Ceramide’s effects on the irradiated germ cells of *C. elegans* have been investigated by Yang et al. [84]. *C. elegans* were grown in normal room conditions for the experiment. Then, the L_4_ worms were treated with ceramide solution for 24 h. The L_4_-staged larvae were irradiated by gamma-ray of 3.37 Gy/min with different doses of 0, 5, 10, 25, and 50 Gy. After irradiation, the worms were collected in groups of 200–300, and the germ cell apoptosis in *C. elegans* was determined. They proposed that DNA damage response in irradiation-induced *C. elegans* can be regulated by a ceramide solution leading to a reduction in the effects of irradiation on germ cell apoptosis [85].

Several studies have been conducted to investigate substances that improve the radiation resistance of *C. elegans*. Very few of these addressed effect of dietary interventions on reducing ionizing irradiation, and most studies focused on reduction of effects of the ultraviolet (UV) irradiation [97,98,99,100,101,102,103,104,105]. However, it would be interesting to know diet can reduce UV irradiation, as the higher energy UV irradiation is also a form of IR [106]. Therefore, numerous studies showed that diets such as didymin (a citrus-derived natural compound having chemical formula C28H34O14, a dietary flavonoid glycoside with antioxidant properties [107]), Leucine, Isoleucine, Valine, arbutin, 2-Selenium-bridged β-cyclodextrin (2-SeCD2- is an enzyme with glutathione peroxidase-like activity), Clostridium butyricum, blueberry, S-allyl cysteine, and apple increased UV irradiation resistance in *C. elegans* [97,98,99,100,101,102,103,104,105]. The effect of these diets on reducing the effect of ionizing irradiation still requires further investigations.

Very few studies investigated effects of dietary interventions on ionizing irradiation. Vitamin A (Vit A), which is a scavenger of radiation products, and polyunsaturated fatty acids (PUFA), which are important cell membrane component, are key elements in the relative fluidity of cell membranes that ensures the normal physiological functioning of cells [108,109]. Liu et al. assessed the protective effects of these supplements against radiation damage in cells. Pretreating the *C. elegans* with Vit A and PUFA notably increased the survival rate and fertility of the worms. Moreover, the shape of the eggs of *C. elegans* changed due to gamma irradiation in a control group without Vit A and PUFA treatment; this experiment showed that using Vit A and PUFA can prevent these effects. Vit A and PUFA also decreased the DNA damage to *C. elegans* in 60Co γ-radiation [109]. Other studies showed the increase life span of *C. elegans* by the reduction in the level of oxidative stress [110,111]. Moreover, resveratrol, which is a strong radical scavenger that regulates superoxide dismutase (SOD) expression, protected against hazardous ionization radiation by reducing oxidative stress. Kan et al. showed that the pre-treatment of *C. elegans* with a dietary intake of a trace amount of resveratrol increased the mean life duration of γ-ray-irradiated *C. elegans* by preventing oxidative stress [112]. Chen et al. also demonstrated that resveratrol’s effect on reducing oxidative stress in *C. elegans*, ultimately increased life span [113]. Hence, only very few studies on dietary or pharmacological intervention to reduce the impact of acute and chronic exposure to irradiation have been conducted. Moreover, effects of these interventions to acute and chronic exposure, separately has also not been discussed thus far. Therefore, studies on dietary or pharmacological intervention to reduce effects of various radiation types and doses in *C. elegans* would contribute advancing the knowledge in radiation biology of the worm and potential translation to animals and humans.

## 5. Conclusions

The use of *C. elegans* as a model organism for biological experiments has increasesed due to its ease of culture and significant genetic homology with humans. Various experiments have been conducted by irradiating *C. elegans* with different radiation types to investigate ultraviolet exposure, X-ray exposure, proton beam exposure, gamma-ray exposure, and β particle exposure. We have grouped these experiments into two sets that analyze chronic irradiation and acute irradiation. *C. elegans* have also been used in space experiments. One of the most important subjects in space experiments on *C. elegans* is resistance to irradiation, as astronauts are constantly vulnerable to space radiation. It is important to study the effects of both chronic and acute radiation on *C. elegans* in space because both irradiation types (acute and chronic) are common there. Solar particle events can be modeled as acute irradiation, while galactic cosmic rays can be considered as chronic irradiation. Based on our review of available literature, we conclude that *C. elegans* are more vulnerable to acute irradiation than chronic irradiation. Acute irradiation decreased life duration and increased the death ratio of *C. elegans*. It also caused severe damage to the DNA of *C. elegans*, which leads to genetic mutations and less reproduction of *C. elegans*. Chronic irradiation also shortened life span and damaged DNA of *C. elegans* compared with control groups, but the effect of chronic irradiation was smaller than that of acute irradiation.

Researchers have also investigated ways to increase the irradiation resistance of *C. elegans*. One approach was by adding specific food supplements such as vitamins, fatty acids and extracts of fruits to the *C. elegans*’ diet. These studies showed that didymin, leucine, valine, arbutin, vitamin A, Clostridium butyricum, S-allyl cysteine, blueberry extract, and apple extract strengthened the resistance of *C. elegans* to irradiation from UV rays as well as increase their life span after irradiation. However, their effect on *C. elegans* which were irradiated with ionizing radiation require further investigation. Moreover, a few studies on using dietary interventions to reduce the effect of ionizing radiation showed that vitamin A (Vit A), polyunsaturated fatty acid (PUFA), and a resveratrol rich food source promoted resistance of *C. elegans* to ionizing radiation and increased their life span after irradiation. There is also a wide range of fields that can be studied in the future as this topic warrants further more experimentations. For instance, to date, there is no pharmacological research being conducted on *C. elegans* to mitigate the effects of irradiation in space travel.

Moreover, to our knowledge, no studies have addressed the effects of gamma-ray exposure on humans at different age groups during evacuation after nuclear accidents, using *C. elegans* as a model organism; yet this can be performed by exposing *C. elegans* to gamma irradiation at different ages and modeling the absorbed dose for different exposure geometries. Future experiments would also help determine the biological impact of radiation on aging and its reversal or reduction by dietary bioactive compounds, is a novel and promising research thrust.

## Figures and Tables

**Figure 1 cells-10-01966-f001:**
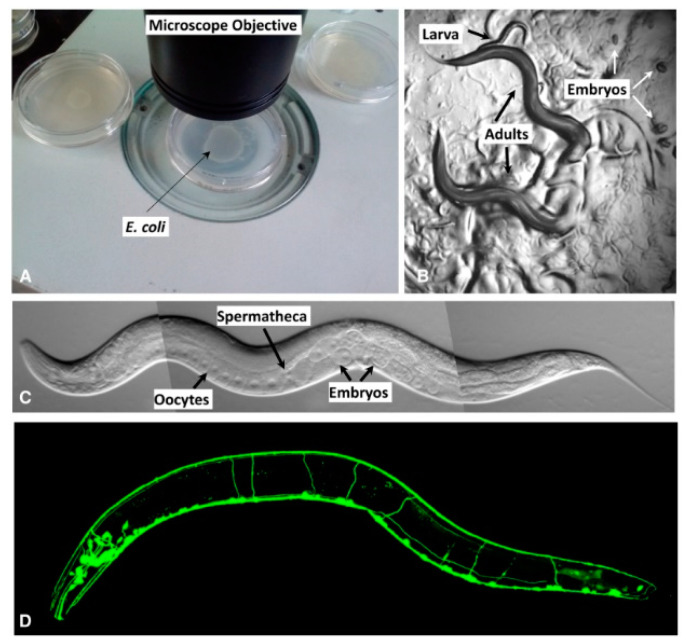
Different views of *C. elegans* in the laboratory. (**A**) Petri dishes with bacterial lawns on the surface of the agar (not visible in this picture). (**B**) Two adults move under a dissecting microscope. (**C**) View from a compound microscope of an adult hermaphrodite. (**D**) The nervous system is shown in the fluorescent image. Photo reprinted with permission of the Genetics Society of America [21,22].

**Figure 2 cells-10-01966-f002:**
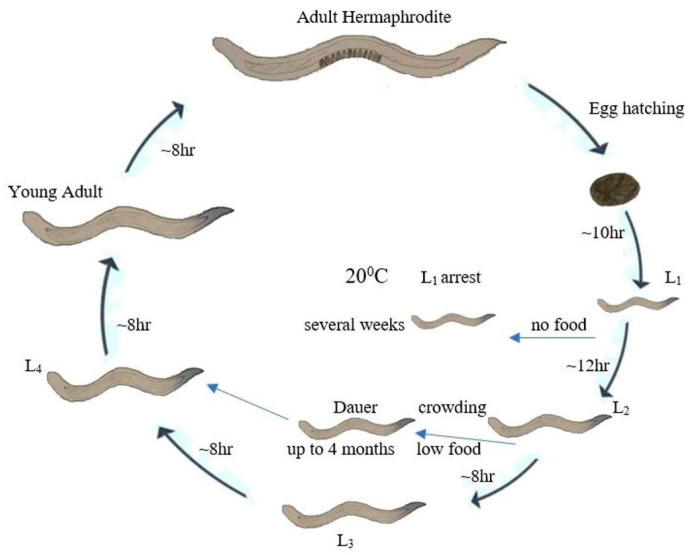
The life cycle of *C. elegans*. Under normal laboratory conditions, *C. elegans* goes through four larval stages (L_1_ to L_4_) in approximately 35 h, and L_4_ larvae molt into adults, which survive for approximately three weeks. Under unfavorable conditions (low food or crowding), L_1_ larvae may proceed through an alternative dauer pathway in which dauer is capable of living for few months and matures to adulthood when the conditions are favorable and proceed to a normal life cycle.

**Figure 3 cells-10-01966-f003:**
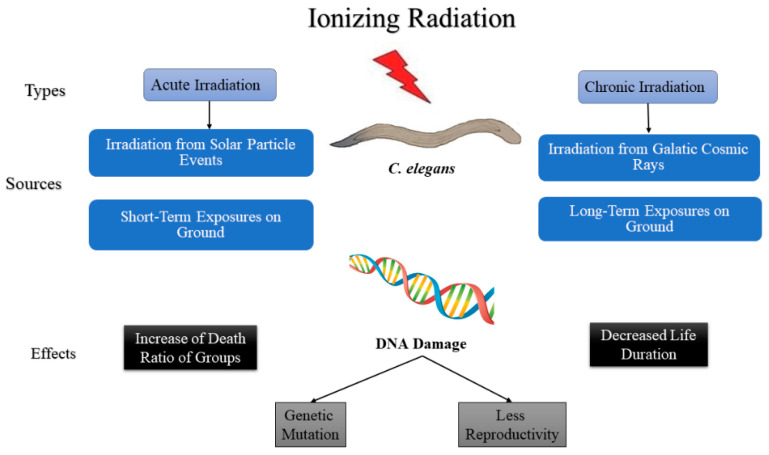
Diagram describing the effect of ionizing radiation on *C. elegans*.

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
