# Peer review of "Review of Biological Effects of Acute and Chronic Radiation Exposure on Caenorhabditis elegans"

_cells, 2021, doi:10.3390/cells10081966_

Round 1

Reviewer 1 Report

This is an interesting review paper on the chronic and acute radiation effect on the C. Elegans radiobiology model. I’m of the opinion that the paper is well-written, comprehensive and clear. I’ve listed some suggested and comments for minor changes below.

L21: Typo - … which show that vitamin A, …

L30: Medical radiation exposure is not only limited to radiotherapy (e.g. diagnostic exposures)? Radiation workers? As we know, the target in radiotherapy is the tumour (much higher doses of 60-65 Gy), so I guess the authors are referring to the dose to the surrounding normal tissue? It might be good to explicitly add this to the text?

L70: I think there is a word missing here? Balancing lethal mutations perhaps?

L112: I guess the authors want to state here that these parameters like temperature, nutrition, reproduction… might impact the worm’s response after radiation exposure and contribute to variations between experiments? Is this causing a bias which should be taken into consideration?

L189: Suggested change:

  • The physical dosimetry was performed with TLDs, a versatile passive dosimeter, which recorded a total dose of 0.8 to 1.1 mGy…
  • Next to this, the conclusion on [56] is a bit confusing, no significant difference is reported on L187, but L191-193 seems to indicate something else?

General comment: What is the sensitivity of this in vivo system? What are the lowest doses which were able to induce significant changes compared to unirradiated worms? Is it proven to be sensitive to doses as low as 1 mGy or are the worms quite radioresistant?

L230: An equivalent dose of … mSv?

L310: The second part of this sentence doesn’t read well, please rephrase.

L335: It might be difficult to find this information in some publications, but if feasible, please try to report the doses in a uniform way (preferably absorbed dose in Gy…) to make it easier to compare studies?

L339: This is the hatching rate for the eggs that survived after 4 hours? Perhaps try to make this clear in the text?

L355: Briefly describe what the role is of HUS-1 is in DDR, similar to what the authors nicely did for Rad-51 a couple of lines higher up.

L420-421: It might be good to refer to Table 1 and 2 at the beginning of Section 3, so the reader is more urged to check the structured overview in the tables while reading section 3.

Table 1 and 2: Some gamma-ray sources are defined (Cesium-137) while other sources are not defined. I guess this information was not available in the papers?

I currently miss some discussion on the following topics, which could perhaps be added to the conclusion part of the paper?

  • Can the authors briefly say something on the impact of radiation quality and dose rate? Were there differences between the dose rate observations for low-LET and high-LET radiation qualities?
  • Are there limitations to the C. Elegans model system?
  • There are quite large differences between the dose rates encountered in the space experiments (mGy-range) compared to the ground-based experiments with much higher doses. What is the sensitivity limit of the model and would it be important to start focussing on low-dose exposure in laboratory experiments?
  • The factors listed under section 2.1 might influence the outcome of the radiobiology experiments? How does this affect the studies listed in Table 1 and 2? Could this cause an experimental bias (e.g. particularly for the chronic exposures, temperature, oxygen… might be different) which could impact the conclusions?

Author Response

Thank you very much for reviewer to provide very insightful comments on article. This will help us to do improvement on paper. We have attached the point by point responses. 

Reviewer 2 Report

In the manuscript entitled “Review of biological effects of acute and chronic radiation exposure on Caenorhabditis elegans, the authors reviewed the effects of acute and chronic radiation on the model organism C. elegans. Totally, this manuscript is informative and well organized. However, we have two small points the authors should consider: 1) The part 2 “Life cycle” should not be very necessary, as the readers interested in this article should also know more about the C. elegans. 2) I personally think the tissue model of radiation-induced reproductive cell death established by J.B. Weidhaas and its related radiation effects should be in this review.

Author Response

Thank you very much to reviewer for providing very insightful comments and suggestion for the improvement of article. Please see the attachment. 
